# Comprehensive Evaluation of Land Use Benefit in the Yellow River Basin from 1995 to 2018

**Zehui Chen** [1], **Qianxi Zhang** [1], **Fei Li** [1,2,3,*] **and Jinli Shi** [1]

[1] Department of Urban and Environmental Sciences, Northwest University, Xi'an 710127, China; 2018116110@stumail.nwu.edu.cn (Z.C.); 2018116118@stumail.nwu.edu.cn (Q.Z.); 2018116133@stumail.nwu.edu.cn (J.S.)

[2] Yellow River Institute of Shaanxi Province, Xi'an 710127, China

[3] Shaanxi Key Laboratory of Earth Surface System and Environmental Carrying Capacity, Xi'an 710127, China

[*] Correspondence: lifei@nwu.edu.cn; Tel.: +86-029-8830-8415

**Abstract:** Land resources are the basis of human survival and development. Land use benefit is the embodiment of land input-output ability. As an important economic zone and ecological barrier in China, it is important to calculate the land productivity in the Yellow River Basin. Using the center of gravity model and other methods, this study evaluated the land use benefit of the Yellow River Basin from 1995 to 2018 based on the selected indicators of geographic grid-scale and analyzed the regional disparity. The results revealed that the comprehensive benefits, economic benefits, and social benefits of land use were on the rise, but the ecological benefits changed in volatility. Land circulation had a great impact on the change of land use benefits. So reasonable land transfer policy should be particularly significant for land use in the Yellow River Basin. In addition, there were obvious spatial differences and agglomeration effects in land use benefit. The high values of benefits were concentrated in urban groups, which showed that areas with better economic and social development had better land use benefits. To narrow land use benefits' spatial differences between regions, the less developed areas deserve more preferential policies to improve their economic and social levels. Besides, ecological benefits are generally not high. Thus, the land policy in the Yellow River Basin should take ecological priority as the basic principle while considering economic factors.

**Keywords:** land use; land use benefit; land circulation; center of gravity model; the Yellow River Basin

## 1. Introduction

Land is a complex system consisting of an economy, society, and ecological environment [1]. As a carrier of human activities, land is the most basic natural resource and material base for human survival and development [2]. Effective land use is therefore an important guarantee for sustainable development. Since the reform and opening up, the process of industrialization and urbanization has accelerated, and the contradiction between man and land has intensified. The traditional extension of land use has led to competition among grain production, urbanization, and ecological environment construction on land resources, which has further led to the contradiction between supply and demand of land. Now China is in a period of rapid socio-economic development, so how to alleviate a series of land problems caused by extensive land abuse is one of the most important prerequisites for achieving high-quality development.

As an important indicator for measuring the sustainability and rationality of land resource utilization, the land use benefit has been one of the hottest issues in recent years. Land use benefit is the material output and effective value of the resources and labor capital invested in the area per unit area of land [3]. The higher the land use benefits, the more reasonable the arrangement of land resources in terms of quantitative structure and space allocation, the higher the level of intensive land use, and the greater the potential for sustainable development. In the studies about land use benefit at home and abroad, the

research perspectives have gradually changed from single benefit evaluation to compre-hensive evaluations of economic, social, and ecological benefits; the research methods have been gradually diversified and complicated, including the entropy weight TOPSIS method, the two-dimensional quadrant method, the matter-element model, the Delphi method, and the time series multi-index method [4–7]; the research contents have involved benefit eval-uation, spatial-temporal evolution, and coupling coordination degree [8–14]; the research scale has also expanded from micro to macro, including county, city, province, basin, urban agglomeration, and even the whole country; the research area has evolved from a simple administrative division to a special land or area with certain characteristics [15–18].

Overall, the existing researches have realized the quantitative evaluation of land use benefits to some extent. Most scholars' researches were mainly through the establishment of a land use benefit evaluation index system. However, the selection of indicators tended to focus solely on economic and social output, without considering changes in the land itself and the benefits of different types of land use. Different types of land's material outputs and effective values are likely to be different. Changing the land use types can result in changes in land use benefits. These should not be ignored in assessing benefits. Additionally, most of the research scales were still relatively single, only focusing on administrative division units. However, administrative unit research data are readily impacted by administrative factors. In addition, due to the influences of different natural and social conditions, inevitably, the data type framework is not unified, which is not conducive to the integration of natural and cultural data [19]. The researches on micro grid-scale remain to be innovated. Therefore, this study added the land use status factor in the selection of the index and selected the geographic grid-scale, trying to solve the following two scientific problems: (1) How to assess land use benefit and discuss its change from the angle of land use pattern; (2) How to break through the restriction of the administrative division unit, realize the integration of natural and cultural data, and observe the spatial heterogeneity of land use benefit from the perspective of the whole region.

Moreover, when choosing research regions, most researchers preferred to choose certain regions with good economic foundations and a high level of social development as research areas. However, they paid less attention to the poorly-developed regions such as the Yellow River Basin, where social and economic development is relatively backward and the ecological environment system is relatively fragile. However, the current state of land use in these underdeveloped regions may be more worrisome and require more scientific evaluation and improvement. According to the 2019 Forum on Ecological Protection and Quality Development of the Yellow River Basin in China, the Yellow River Basin played an important role in China's economic and social development and ecological security and had a great significance in ecological protection and high-quality development [20]. As an important economic region and ecological barrier in China, the Yellow River Basin has nearly a third of the country's land resources and populations, but the overall quality of development is not high. So, it is of great significance to scientifically evaluate the land use benefit of this region and formulate targeted land resource utilization methods and policies.

Therefore, this study took the Yellow River Basin as the study area and comprehen-sively analyzed the temporal and spatial variation characteristics of land use benefits in this basin from 1995 to 2018, selecting specific measurement indicators from the three dimensions (economic benefit, social benefit, and ecological benefit). It can provide some references and support for the utilization and management of land resources in the Yellow River Basin.

## 2. Materials and Methods

### 2.1. Study Area

The Yellow River Basin (Figure 1) originates in Bayan Har Mountain, Qinghai Province, which spans 1900 km from east to west and 1100 km from north to south and flows through Qinghai, Sichuan, Gansu, and another nine provinces with a total area of 795,000 km$^2$ [21]. By the end of 2018, the permanent population of the Yellow River Basin was 420 million,

accounting for 30.3% of the total population of the country, and the regional GDP was 23.9 trillion yuan, accounting for 26.5% of the national GDP. The terrain is high in the west and low in the east, and the landform types are diverse. The west is composed of mountains, and the middle is the loess landform. The east is the Yellow River alluvial plain. The main land use types are grassland, cropland, and woodland.

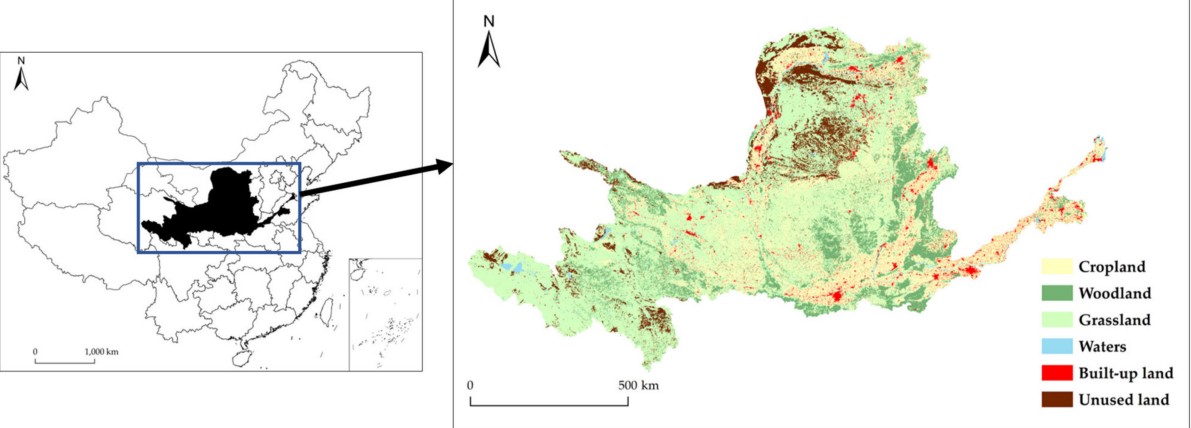

**Figure 1.** Map of land use types in the Yellow River Basin (2018).

## 2.2. Data Sources

The data used in this study mainly include land use data, socio-economic data, and meteorological data. The land use data in 1995, 2000, 2005, 2010, 2015, and 2018 came from the Land Use Remote Sensing Monitoring Database of China at the Resource and Environmental Data Cloud Platform (http://www.resdc.cn (accessed on 15 June 2021)). The socio-economic data for 1995–2018 were from the 1995–2018 China Statistical Yearbook (https://data.cnki.net/ (accessed on 15 June 2021)); the Yellow River Yearbook; the statistical yearbooks of the provinces in the Yellow River Basin; and the GDP spatial distribution kilometer grid dataset and population spatial distribution kilometer grid dataset of China at the Resource and Environmental Data Cloud Platform (http://www.resdc.cn (accessed on 15 June 2021)), including GDP, population density, per capita income, Engel Coefficient, and proportion of the urban population. The climate data were downloaded from the Resource Environmental Science and Data Center, Chinese Academy of Sciences (http://www.resdc.cn (accessed on 15 June 2021)), and included spatial interpolation data of annual precipitation and annual mean temperature.

## 2.3. Study Methods

Using the ArcGIS platform Create Fishnet tool to create a 10 km × 10 km grid, a total of 8629 geographic grids with an area of 100 km$^2$ were obtained. The grid-scale aimed to reflect the spatial difference of land use benefit more accurately at the micro-scale. The grids were superimposed with the land use data of 1995, 2000, 2005, 2010, 2015, and 2018, and the key core indicators were selected. The land use benefit evaluation values from 1995 to 2018 were calculated based on different land use types in the grids.

### 2.3.1. Land Use Economic Benefit

Economic outputs of land use are mainly GDP. Therefore, this study constructed a model combining GDP with land use status to characterize the land use economic benefits in the Yellow River Basin. The model's formulas constructed in this study are as follows:

$$I_{G=} \sum_{k=1}^{6} VG_k \times A_{nk} \tag{1}$$

$$VG_k = \frac{\sum_{n=1}^{6} GDP_{dnk}}{6} \qquad (2)$$

$I_G$ represents the land use economic benefit (yuan). $A_{nk}$ represents the area of the k-th type of land utilization in the $n$-th year (km$^2$). $VG_k$ represents the economic benefit's value per unit area of the k-th type of land use (yuan/km$^2$). $GDP_{dnk}$ represents the GDP per unit area of the k-th type of land use in the $n$-th year (yuan/km$^2$). The economic benefit value per unit area was calculated by Formula (2) (Table 1).

**Table 1.** Value of economic benefits per unit area of the Yellow River Basin from 1995 to 2018.

| Economic Benefit Value per Unit Area (yuan/km$^2$). | | | | | |
|---|---|---|---|---|---|
| Cropland | Woodland | Grassland | Waters | Built-up Land | Unused Land |
| 9,408,367.72 | 3,652,102.03 | 2,384,930.41 | 8,101,054.62 | 44,470,365.73 | 1,531,455.38 |

### 2.3.2. Land Use Social Benefit

The social benefits of land use are mainly reflected in regional social development and people's living level. Population density and per capita income are often used as indicators to measure this level. So, population density, per capita income, and land use condition were merged in this study to characterize the social benefits of land use in the Yellow River Basin. The model's formulas constructed in this study are as follows:

$$I_{P=} \sum_{k=1}^{6} VP_k \times A_{nk} \qquad (3)$$

$$VP_k = \frac{\sum_{n=1}^{6} POP_{dnk} \times RJ_n}{6} \qquad (4)$$

$I_P$ represents the land use social benefit (yuan). $VP_k$ represents the social benefit's value per unit area of the k-th type of land use (yuan/km$^2$). $POP_{dnk}$ represents the population density per unit area of the k-th type of land use in the $n$-th year (person/km$^2$). $RJ_n$ represents the per capita income of the Yellow River Basin in the $n$-th year (person/yuan). The social benefit value per unit area was calculated by Formula (4) (Table 2).

**Table 2.** The value of social benefits per unit area of the Yellow River Basin from 1995 to 2018.

| Social Benefit Value per Unit Area (yuan/km$^2$) | | | | | |
|---|---|---|---|---|---|
| Cropland | Woodland | Grassland | Waters | Built-up Land | Unused Land |
| 2,853,090.92 | 1,200,556.97 | 711,703.41 | 1,830,832.29 | 7,935,354.21 | 243,866.86 |

### 2.3.3. Land Use Ecological Benefit

The ecological benefits of land use are mainly reflected in the ecological environment. Previous studies have selected "forest coverage", "sewage treatment rate" and other indicators to be evaluated [22,23]. However, there are many factors that affect a regional ecosystem, such as land, climate, precipitation, etc. Several single indicators are difficult to reflect the ecosystem environment of the whole region. Therefore, this study used ecosystem service value to characterize the ecological benefits of land use in the Yellow River Basin. Ecosystem services are life-support products and services obtained directly or indirectly through ecosystem structures, processes, and functions [24–28], which are important indicators of a regional ecosystem. Based on the research results of Costanza and Xie Gaodi et al. [29,30], the land use types were reclassified (Table 3). This study mainly referred to the model of Li Fei et al. [31], adjusting the biomass factors and socio-

economic factors of the ecosystem service value per unit area in China according to the actual conditions of the Yellow River Basin.

$$I_{E=} \sum_{k=1}^{6} ESV_k \times A_{nk} \tag{5}$$

$$ESV_k = \sum_{i=1}^{9} VC_{ki} \times S \times PI \tag{6}$$

**Table 3.** Correspondence of land use types.

| Types of Land Use in this Research | Types of Land Use in Reference [29] |
|:---:|:---:|
| Cropland | Cropland |
| Woodland | Forest |
| Grassland | Grassland |
| Waters | Wetlands, rivers/lakes |
| Built-up land | - |
| Unused land | Deserts |

$I_E$ represents the land use ecological benefit (yuan). $ESV_k$ represents the ecological benefit's value per unit area of the k-th of land use (yuan/km$^2$). $VC_{ki}$ represents the ecosystem service value of the k-th type of land use and the i-th type of function per unit area of the Chinese ecosystem (yuan/km$^2$) [29]. $S$ represents the biomass factor adjustment coefficient, and the calculated value is 1.00. $PI$ represents the socio-economic factor adjustment coefficient and the calculated result is 0.77. Ecological benefits calculated per unit area (Table 4) by Equation (6).

**Table 4.** The value of ecological benefits per unit area of the Yellow River Basin from 1995 to 2018.

| Ecological Benefit Value per Unit Area (yuan/km$^2$) | | | | | | |
|:---:|:---:|:---:|:---:|:---:|:---:|:---:|
| **Primary Type** | **Secondary Type** | **Cropland** | **Woodland** | **Grassland** | **Waters** | **Built-Up Land** | **Unused Land** |
| Provision service | Food production | 34,682.63 | 11,445.04 | 14,913.30 | 30,867.62 | 0.00 | 693.50 |
| | Raw material production | 13,526.30 | 103,354.40 | 12,486.06 | 20,462.83 | 0.00 | 1387.00 |
| Regulation service | Gas regulation | 24,971.34 | 149,828.83 | 52,023.95 | 101,273.14 | 0.00 | 2081.27 |
| | Climate regulation | 33,642.39 | 141,158.55 | 54,105.22 | 541,396.62 | 0.00 | 4508.51 |
| | Hydrological regulation | 26,705.86 | 141,852.05 | 52,717.45 | 1,117,127.57 | 0.00 | 2428.02 |
| | Disposal of waste | 48,208.94 | 59,653.98 | 45,780.92 | 1,014,467.44 | 0.00 | 9017.79 |
| Support service | Soil conservation | 50,983.70 | 139,424.04 | 77,688.79 | 83,238.32 | 0.00 | 5896.28 |
| | Biodiversity conservation | 35,376.13 | 156,418.60 | 64,856.76 | 246,940.20 | 0.00 | 13,873.05 |
| Cultural service | Aesthetic landscape | 5896.28 | 72,140.03 | 30,174.12 | 316,652.22 | 0.00 | 8323.52 |
| | Sum | 273,993.58 | 975,275.52 | 404,746.57 | 3,472,425.96 | 0.00 | 48,208.94 |

The calculation formula of $S$ is as follows:

$$S = \frac{NPP_h}{NPP_g} \tag{7}$$

$NPP_h$ represents the vegetation net primary productivity in the Yellow River Basin (t/ha/a). $NPP_g$ represents the vegetation net primary productivity in China (t/ha/a). The Thornthwaite Memoria model proposed by Lieth et al. was used to estimate the NPP in this study [32,33]. That is, the annual average temperature and annual average precipitation were used to calculate the annual actual evapotranspiration and annual average evapotranspiration in the Yellow River Basin, and further obtain the net primary productivity of vegetation. The calculation model is as follows:

$$NPP = 3000 \left[ 1 - e^{-0.0009695(V-20)} \right] \tag{8}$$

$$V = \frac{1.05 Pre}{\sqrt{1 + \left( \frac{1.05 Pre}{L} \right)^2}} \tag{9}$$

$$L = 3000 + 25 Tmp + 0.05 Tmp^3 \tag{10}$$

$V$ is the annual actual evapotranspiration (mm). $L$ is the annual average evapotranspiration (mm). $Pre$ is the annual precipitation (mm). $Tmp$ is the annual mean temperature (°C).

The calculation formulas of $PI$ are as follows:

$$PI = W \times A \tag{11}$$

$$W = \frac{w_h}{w_g} \tag{12}$$

$$w = \frac{2}{(1 + e^{-m})} \tag{13}$$

$$A = \frac{GDP_{mh}}{GDP_{mg}} \tag{14}$$

$$m = \frac{1}{En_t - 2.5} \tag{15}$$

$$En_t = En_{tr} \times (1 - P_{tu}) + En_{tu} \times P_{tu} \tag{16}$$

$W$ is the willingness to pay for $ESV$ and can be calculated by Logistic regression model. $W_h$ is the willingness to pay of the Yellow River Basin. $W_g$ is the willingness to pay in China. $A$ is the ability to pay. $GDP_{mh}$ is the GDP per capita in the Yellow River Basin (yuan/person). $GDP_{mg}$ is the GDP per capita in China (yuan/person). $m$ is the coefficient of social development stage. $En_t$ is the Engel Coefficient of the Yellow River Basin in t years. $En_{tr}$ is the urban Engel Coefficient of the Yellow River Basin in t years. $En_{tu}$ is the rural Engel Coefficient of the Yellow River Basin in t years. $P_{tu}$ is the proportion of the urban population of the Yellow River Basin in t years [34–36].

### 2.3.4. Jenks Natural Breaks Classification Method

The Jenks natural breaks classification method [37] was used to divide the land use benefit into low, medium, and high value areas. The method is based on the inherent natural grouping in the data [38]. The classification interval is identified, similar values can be grouped in the most appropriate way, and the difference between each class can be maximized. The grouping method is to divide the data into multiple classes, and for these

classes, set their boundaries at positions where the difference in data values is relatively large. The method's formulas are as follows:

$$\overline{X} = \frac{1}{n} \sum_{i=1}^{n} X_i \tag{17}$$

$$SDAM = \sum_{i=1}^{n} \left( X_i - \overline{X} \right)^2 \tag{18}$$

$\overline{X}$ is the mean of a class of arrays. $SDAM$ is the sum of the squares of the total deviation of the array. $n$ is the number of elements in the array. $X_i$ is the value of the i-th element. Calculate the class sum of squared total deviations $SDCM$ for each range combination in the classification result. Find the smallest value among them and record it as $SDCM_{min}$. Divide $N$ elements into $k$ categories, so that the classification result can be divided into $k$ subsets, one of which is $[X_1 X_2 \cdots X_i] \cdots [X_{j+1} X_{j+2} \cdots X_n]$. Then calculate the sum of squared total deviations for each subset $SDCM_i \cdots SDCM_n$. The calculation formula of $SDCM_1$ is as follows:

$$SDCM_1 = SDAM_i + SDAM_j + \cdots + SDAM_n \tag{19}$$

Similarly, the $N$ elements can also be divided into other cases of class $k$, and $SDCM_2 \cdots SDCM_n$ can be calculated in turn. Select the smallest value as the final result as $SDCM_{min}$. So, the classification range is the best classification. The above calculation process can be achieved directly through ArcGIS 10.2.

### 2.3.5. Layer Stacking

Layer Stacking allows a single band raster atlas to be rendered together to create a color composite image. Based on the Layer Stacking function in the ENVI 5.3 software Toolbox, the economic benefits, ecological benefits, and social benefits were respectively combined in the order of red, green, and blue in ENVI 5.3. Adopting this method is not only conducive to the superimposing benefits, but the true color is also convenient for observation.

### 2.3.6. Gravity Center Model

The movement of the geographical center of gravity can reflect the change of geographical elements. So, we used this model to visually analyze the spatial changes of geographic elements in the process of regional development [39,40]. The calculation formulas of the gravity center model (Figure 2) are as follows:

$$\overline{x} = \frac{\sum_{i=1}^{n} M_i x_i}{\sum_{i=1}^{n} M_i} \tag{20}$$

$$\overline{y} = \frac{\sum_{i=1}^{n} M_i y_i}{\sum_{i=1}^{n} M_i} \tag{21}$$

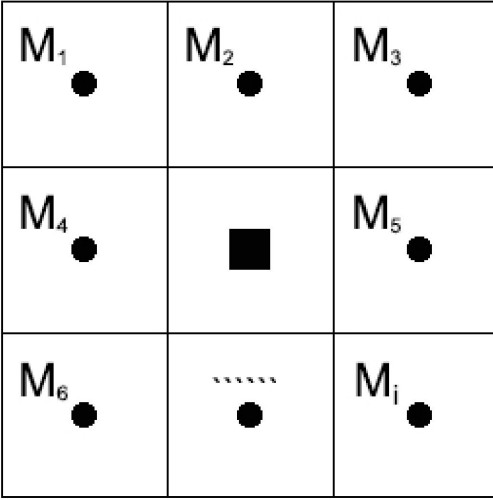

**Figure 2.** Center of gravity model diagram.

$(\overline{x}, \overline{y})$ is the geographical unit coordinates of gravity. $(x_i, y_i)$ is the coordinates of gravity center of the i-th sub-unit; and $M_i$ is the benefit value of sub-units.

### 2.3.7. Getis-Ord Gi*

The Getis-Ord Gi* can detect and distinguish the correlation caused by the aggregation of high and low value regions in space, so we used it to analyze the spatial agglomeration phenomenon of land use benefit.

The formula is as follows:

$$G_i * (d) = \sum_{i=1}^{n} W_{ij}(d)x_i / \sum_{i=1}^{n} x_i \tag{22}$$

$x_i$ is the observation value of area unit $i$. $W_{ij}(d)$ is the space weight. It is 1 if the spaces are adjacent, and 0 if they are not. If the value of $G_i * (d)$ is significantly positive, it indicates that the value around region $i$ is relatively high and belongs to the hot spot area. On the contrary, it is the cold spot area.

## 3. Results

### 3.1. Time Change of Land Use Benefit

Vertically, the comprehensive land use benefits in the Yellow River Basin increased from 58,329.99 hundred million yuan in 1995 to 63,820.87 hundred million yuan in 2018, with an average increase of 9.41% (Figure 3). Economic benefits and social benefits also maintained a steady growth trend with an average increase of 11.06% and 6.16%, respectively. Both benefits grew fastest in 2015–2018 with annual growth rates of 1.91% and 1.12%, respectively. This period was between the end of the 12th 5-Year Plan and the beginning of the 13th 5-Year Plan. A series of macroeconomic policies such as economic structural adjustment and development mode transformation led to the continuous expansion of built-up land in the basin and the constant increase of regional GDP and per capita income. Ecological benefits fluctuated over time with an average increase of 0.76%, with the fastest growth rate of 0.08% per year in 2010–2015. Ecological benefits decreased but then increased in 1995–2000, which was closely related to the project of reforestation and returning pasture land to grass in 2000. In this period, the areas of grassland, water, and other ecological land expanded in the basin, which can improve the ecological environment.

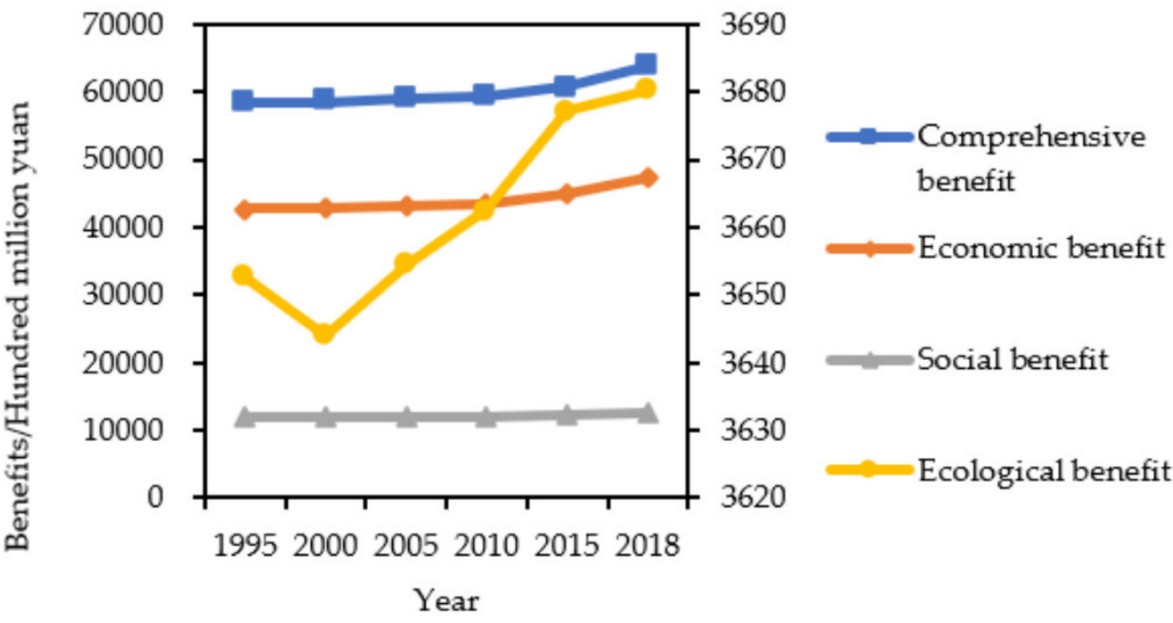

**Figure 3.** Land use benefits of the Yellow River Basin from 1995 to 2018.

Horizontally, the economic benefits rose most rapidly, followed by social and comprehensive benefits. However, the ecological benefits changed little over time from 1995 to 2018. The difference in the growth rate had shown that some development policies significantly had promoted the economic and social development in the Yellow River Basin over the past 20 years. Although some improvements had been made in the construction of ecological civilization, the achievements were not significant compared with those of the economy and society. The reason may be that the ecological environment in the Yellow River Basin is fragile, which leads to difficulty in improvement [41]. In addition, the trend of comprehensive benefits was consistent with that of economic and social benefits, and the growth rates were similar but were quite different from that of ecological benefits. It showed that the changes of economic and social benefits of land use were the main factors leading to the change of comprehensive benefits, but the ecological benefits had little effect on comprehensive benefits.

### 3.2. Spatial Distribution of Land Use Benefit

The comprehensive benefit in the Yellow River Basin showed that the spatial difference was obvious (Figure 4), which presented that the south was higher than the north and the east was higher than the west. This was consistent with the spatial distribution characteristics measured by Xu Hui et al. on the high-quality development level of the Yellow River Basin [42]. High-value areas were small, concentrated in the southern and the northern edge of the Yellow River Basin. The median areas were concentrated in the middle of the Yellow River Basin, which presented a continuous feature. Low-value areas were concentrated in the western and north-central regions in the Yellow River Basin. The extensive distribution of middle or low value areas showed that the comprehensive benefit of land use in the Yellow River Basin was at a low level and the intensive utilization level of land resources was not high, which was one of the factors leading to the low overall development level in the Yellow River Basin. From 1995 to 2018, the areas with high or low comprehensive benefits of land use in the basin gradually decreased. The median regions slowly expanded to the northwest regions.

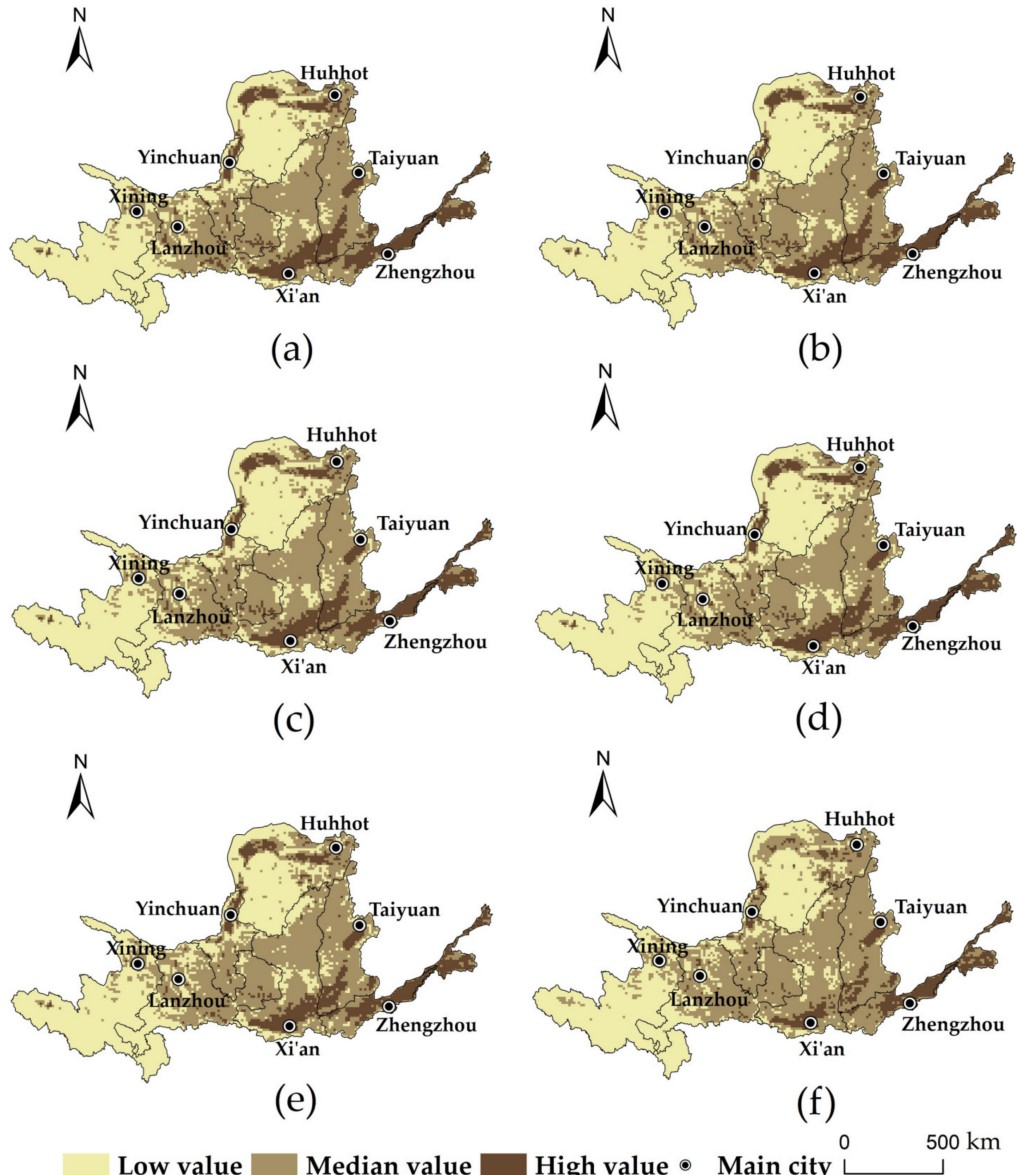

**Figure 4.** Spatial distribution of comprehensive benefits of land use in the Yellow River Basin:
(**a**) 1995; (**b**) 2000; (**c**) 2005; (**d**) 2010; (**e**) 2015; (**f**) 2018.

Low economic benefit areas were widely distributed in the west and north-central
part of the basin (Figure 5). The high-value areas were concentrated in the northern edge
and in the four provinces of Shandong, Henan, Shaanxi, and Shanxi of the southeast.
The middle-value areas were concentrated in the middle of the basin. The distribution
characteristics of social benefits were basically consistent with comprehensive benefits and
economic benefits (Figure 6). The ecological benefits were basically in the median value
areas (Figure 7). The high-value areas gathered in the southeast, and the low-value areas
were mainly distributed in the eastern downstream area and in the middle and north of
the loess plateau area with serious soil erosion.

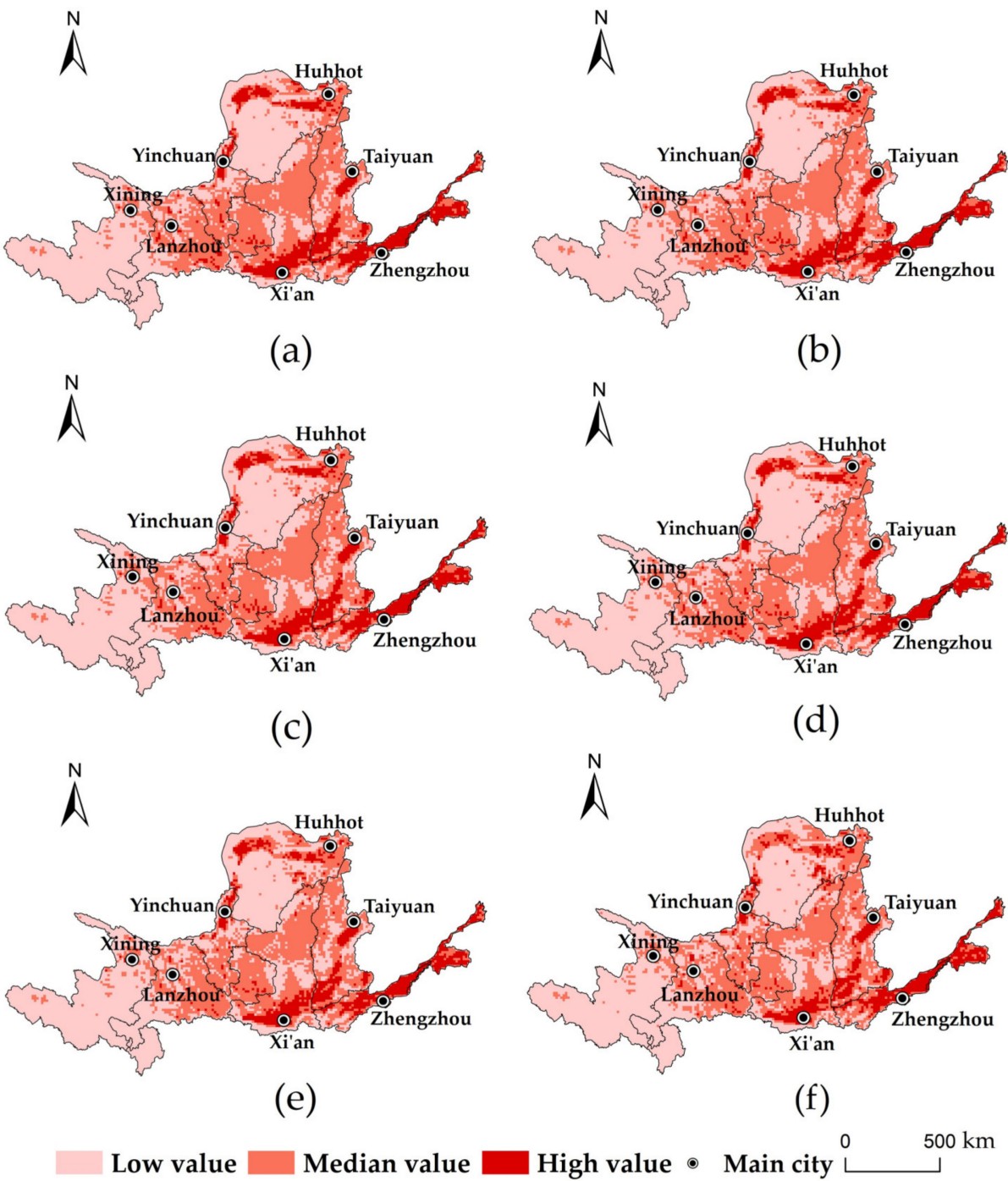

**Figure 5.** Spatial distribution of economic benefits of land use in the Yellow River Basin: (**a**) 1995; (**b**) 2000; (**c**) 2005; (**d**) 2010; (**e**) 2015; (**f**) 2018.

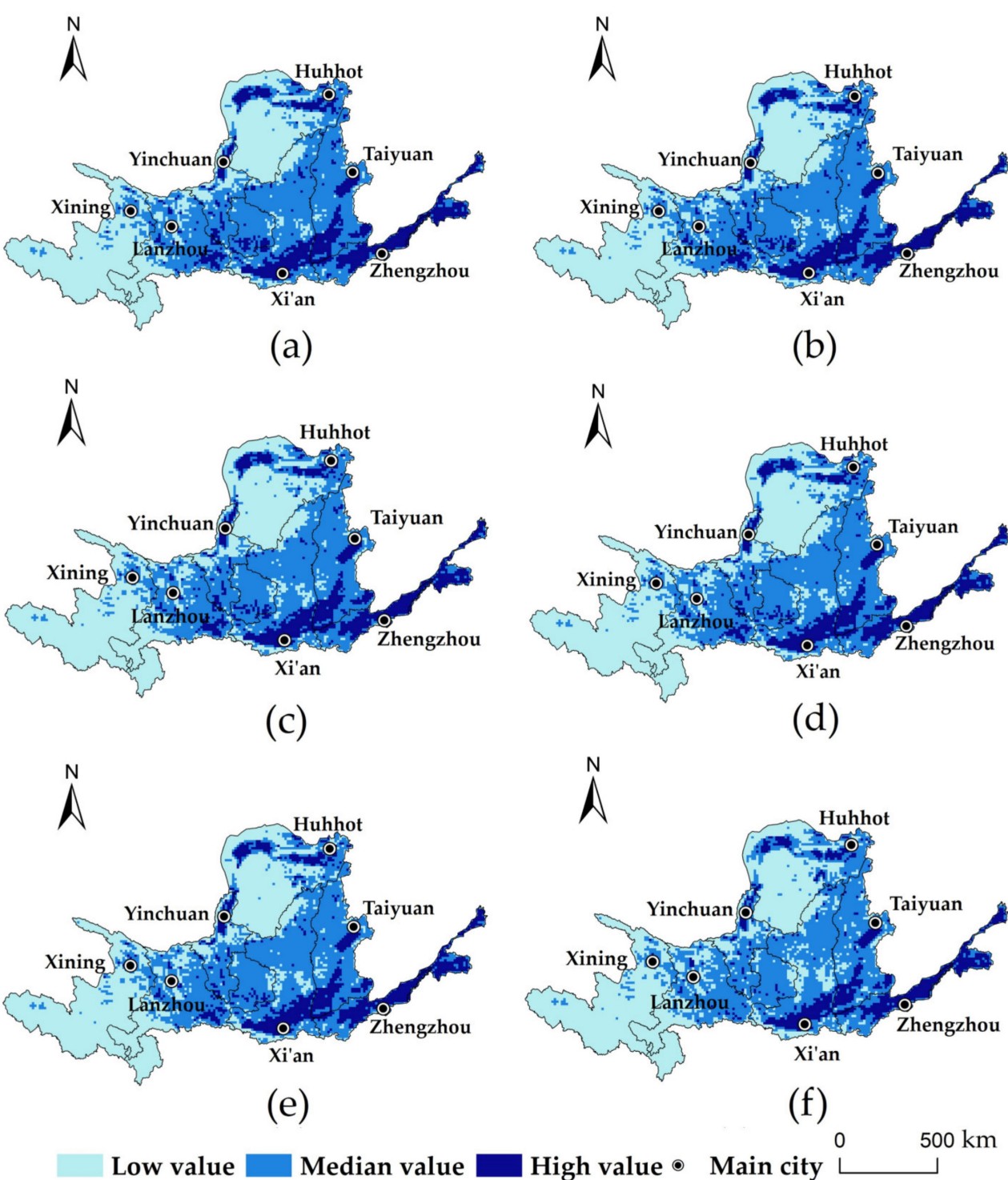

**Figure 6.** Spatial distribution of social benefits of land use in the Yellow River Basin: (**a**) 1995; (**b**) 2000; (**c**) 2005; (**d**) 2010; (**e**) 2015; (**f**) 2018.

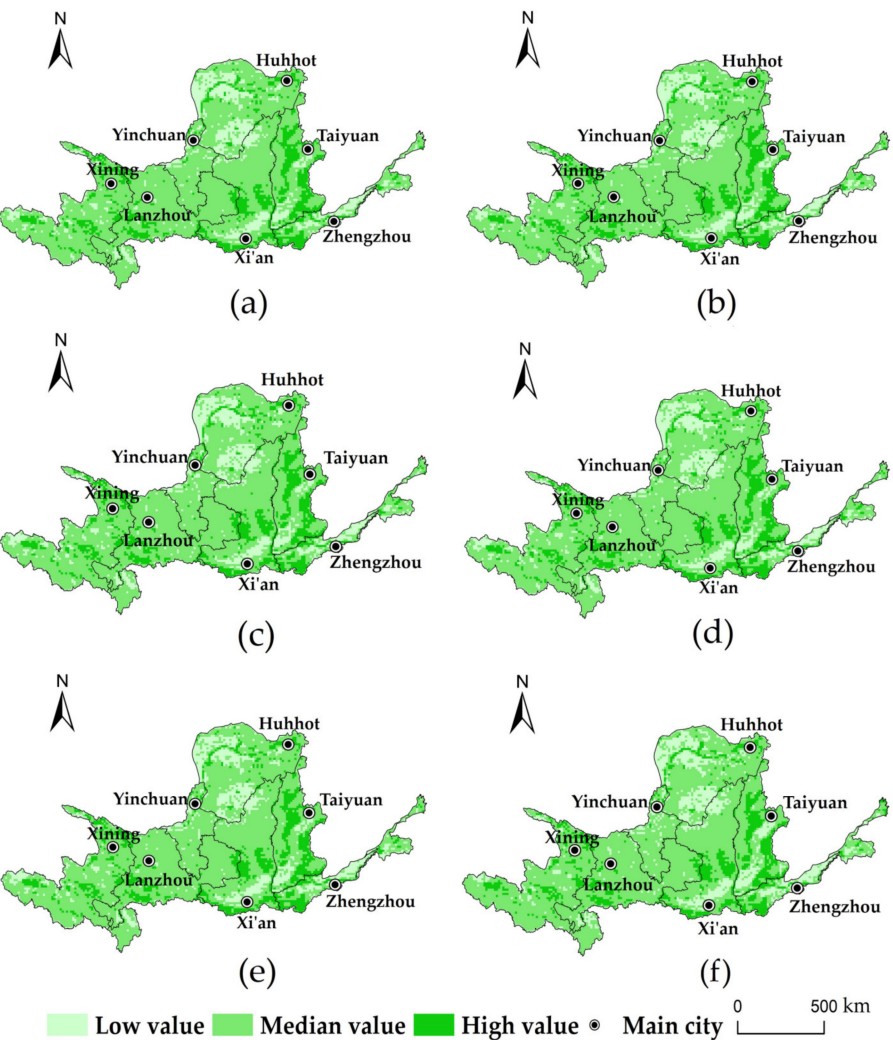

**Figure 7.** Spatial distribution of ecological benefits of land use in the Yellow River Basin: (**a**) 1995; (**b**) 2000; (**c**) 2005; (**d**) 2010; (**e**) 2015; (**f**) 2018.

Using the Layer Stacking, the economic benefit, ecological benefit, and social benefit were respectively combined in the order of red, green, and blue in ENVI 5.3. And according to some basic rules of color superposition, for example, red and blue form purple, and the three colors of red, blue, and green are added to get black. Finally, this study got the following result figure (Figure 8). The result of Layer Stacking showed that the eastern and northern edge areas had bright purple or bright green, indicating that the economic, social, and ecological benefits of land use in these two areas were prominent. The central region had a large area of dark purple distribution, which showed that the economic and social benefit was more prominent than other regions. Green was widely distributed in the western region, which reflected the prominent ecological benefits. The distribution of the above colors further proved that high benefits were concentrated in the eastern and northern edge areas of the Yellow River Basin. At the same time, it reflected the poor coordination between the benefits of the two areas and the obvious polarization of the benefit distribution. Comparative analysis showed that the overall color in the basin became darker from 1995 to 2018, which meant that the polarization distribution among the benefits of land use was alleviated.

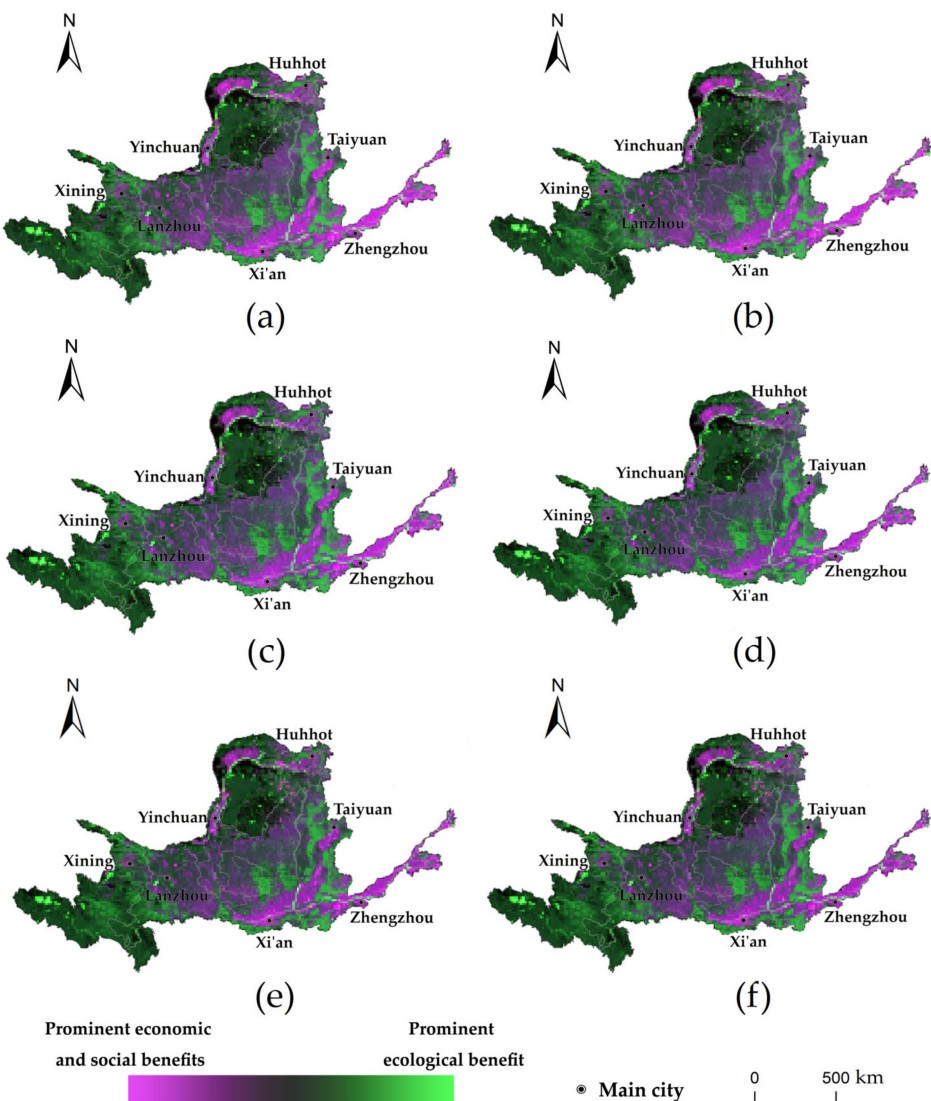

**Figure 8.** The superposition of economic, social and ecological benefits of land use in the Yellow River Basin: (**a**) 1995; (**b**) 2000; (**c**) 2005; (**d**) 2010; (**e**) 2015; (**f**) 2018.

*3.3. Changes in Land Use Benefit's Center of Gravity*

The movement of the geographical center of gravity can reflect the change in dominance of the land use benefits obtained. From 1995 to 2018, the economic, social, and ecological benefits' center of gravities of land use were all in the central region of the Yellow River Basin (Figure 9). The center of gravities of economic and social benefits moved to the northeast, with average speeds of 3 km/year and 1.73 km/year respectively. The center of gravities of ecological benefits moved left in 1995–2000, right in 2000–2010, and left in 2010–2018. Compared with 1995, the center of gravity of ecological benefit moved slowly to the southwest, with an average speed of 1 km/year. By observing the land use benefit's gravity center triangle (Figure 10), it moved slightly to the southeast from 1995 to 2005, then gradually moved to the northwest from 2005 to 2018. Finally, the entire barycenter triangle in 2018 was located above that in 1995, which reflected the slow northward movement of the benefits' high-density areas from 1995 to 2018. This is because the governance of the Yellow River and the development and construction of the poor areas in the upper and middle reaches of the Yellow River promote the gradual shift of the socio-economic focus from the east to the north, which reduce the big regional development differences.

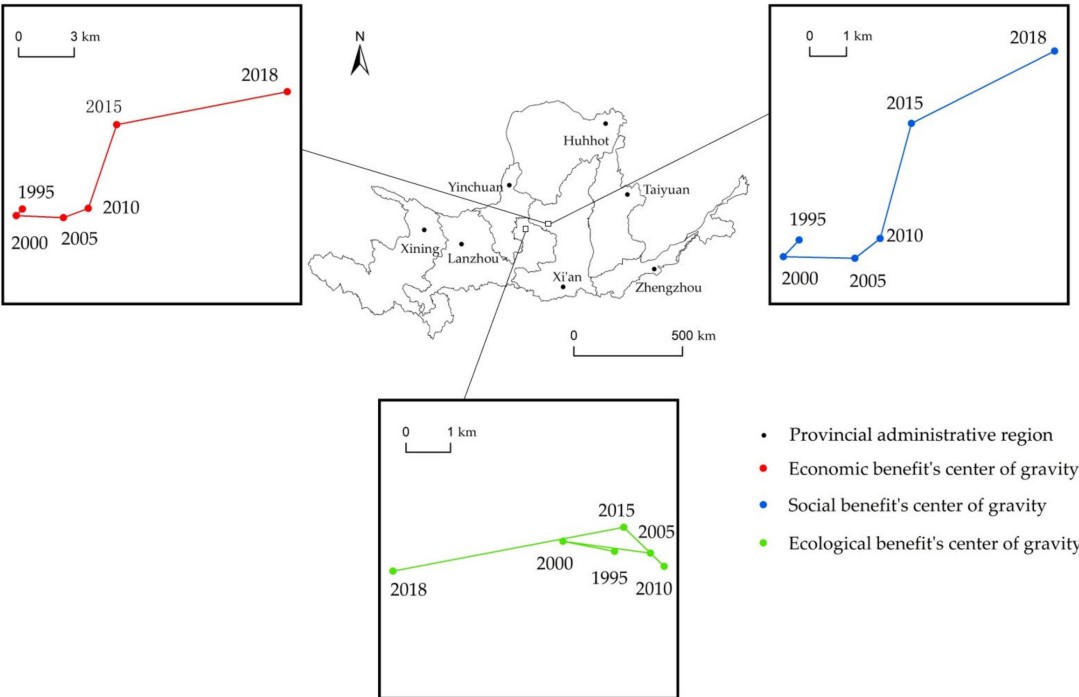

**Figure 9.** The shifts of the center of gravities of land use benefits in the Yellow River Basin from 1995 to 2018.

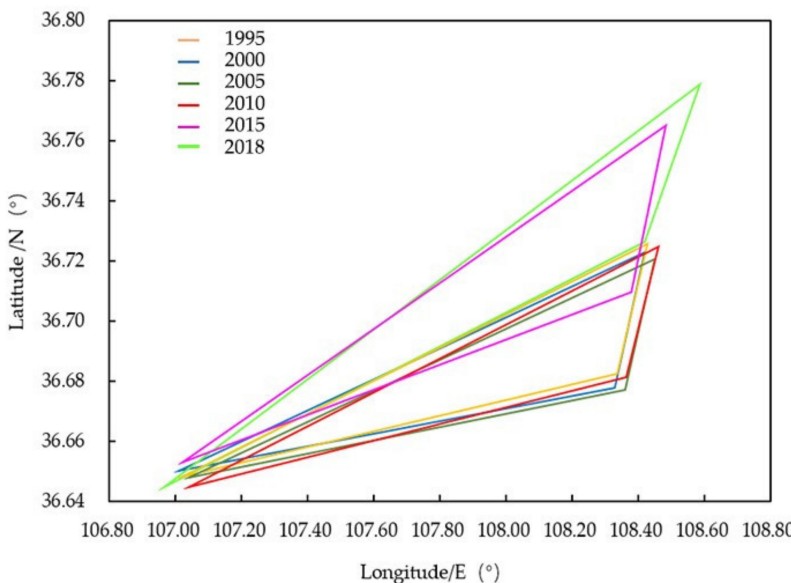

**Figure 10.** The triangles of gravities of land use benefits in the Yellow River Basin from 1995 to 2018.

## 4. Discussion

### 4.1. The Influence of Land Circulation on the Temporal Change of Land Use Benefit

Land circulation will affect land use benefits [43]. The changes of land use type will bring about the changes of land use benefit. The conversion of other types of land use to built-up land was the primary land use reason for the increase in land use economic benefit of the Yellow River Basin (Table 5). Built-up land is the main carrier of economic activities, and the economic growth range is consistent with the expansion degree of built-up land [44]. Therefore, appropriate expansion of built-up land is conducive to the economic development in the basin. The conversion of grassland to cropland was one of the reasons for increasing land use social benefit (Table 6). As the main agricultural area in China, the Yellow River Basin is the focus of the origin and development of dry farming

agriculture in the north [45]. According to the Yellow River Yearbook, the agricultural output value in 2018 accounted for about 60% of the total output value of agriculture, forestry, animal husbandry, and fishery. This showed that agricultural production is not only one of the main social productive activities, but also one of the main sources of per capita income and one of the pillars of social stability and development for the people in the basin. The climate conditions of drought and water shortage [46], poor grassland quality [47], and other factors in the Yellow River Basin make it difficult for animal husbandry and fishery to develop on a big scale. Therefore, we can consider transforming part of grassland into cropland and increase the capital investment in the agricultural sector. This would improve the income security of farmers, rise the enthusiasm of farmers, and enhance the quality of cultivated land. Additionally, the main reason for the increase of land use ecological benefits was the conversion of other types of land use to waters (including wetlands, rivers, and lakes) (Table 7). River and lake play an important role in regulating temperature and stabilizing local climate. Wetland, as the lung of the earth, has the function of water conservation and purification. They are all ecological conservation lands [48]. The protection and return of natural ecological land can effectively protect the ecological services of land, help balance the land ecosystem, and improve the land use ecological benefits [49]. All in all, land use benefits showed increasing trends, but ecological benefits were far lower than economic and social benefits. So, the land use pattern in the Yellow River basin should adopt the ecological priority principle, which would only increase the ecological land. The total amount of cropland remained unchanged and optimized in spatial position [50]. This promotes high quality development and intensive use of built-up land. However, the local government should avoid unreasonable expansion of land. Unreasonable expansion can easily lead to the decoupling between built-up land and population and economic growth, which is not conducive to improving land use benefit [51].

**Table 5.** Land use economic benefit change caused by land use change (hundred million yuan).

| | 2018 | | | | | |
| --- | --- | --- | --- | --- | --- | --- |
| **1995** | **Cropland** | **Woodland** | **Grassland** | **Waters** | **Built-up Land** | **Unused Land** |
| **Cropland** | 0.00 | −754.07 | −3975.27 | −43.14 | 5364.49 | −149.66 |
| **Woodland** | 650.46 | 0.00 | −343.40 | 22.24 | 489.82 | −19.09 |
| **Grassland** | 3729.45 | 359.88 | 0.00 | 188.63 | 2356.78 | −175.82 |
| **Waters** | 39.22 | −17.80 | −160.05 | 0.00 | 218.22 | −52.56 |
| **Built-up land** | −2769.90 | −163.27 | −841.71 | −145.48 | 0.00 | −42.94 |
| **Unused land** | 252.06 | 27.57 | 214.22 | 72.27 | 386.45 | 0.00 |

**Table 6.** Land use social benefit change caused by land use change (hundred million yuan).

| | 2018 | | | | | |
| --- | --- | --- | --- | --- | --- | --- |
| **1995** | **Cropland** | **Woodland** | **Grassland** | **Waters** | **Built-Up Land** | **Unused Land** |
| **Cropland** | 0.00 | −216.48 | −1212.03 | −33.73 | 777.59 | −49.58 |
| **Woodland** | 186.74 | 0.00 | −343.40 | 22.24 | 489.82 | −19.09 |
| **Grassland** | 3729.45 | 138.83 | 0.00 | 36.93 | 404.52 | −96.37 |
| **Waters** | 30.67 | −2.52 | −31.34 | 0.00 | 36.63 | −12.70 |
| **Built-up land** | −401.50 | −26.94 | −144.47 | −24.42 | 0.00 | −7.69 |
| **Unused land** | 83.50 | 12.44 | 117.43 | 17.46 | 69.22 | 0.00 |

**Table 7.** Land use ecological benefit change caused by land use change (hundred million yuan).

| 1995 | 2018 | | | | | |
|---|---|---|---|---|---|---|
| | Cropland | Woodland | Grassland | Waters | Built-up Land | Unused Land |
| Cropland | 0.00 | 91.87 | 74.01 | 105.55 | −41.92 | −4.29 |
| Woodland | −79.24 | 0.00 | −154.61 | 12.49 | −11.70 | −8.34 |
| Grassland | −69.43 | 162.03 | 0.00 | 101.23 | −22.67 | −73.45 |
| Waters | −95.95 | −9.99 | −85.90 | 0.00 | −20.83 | −27.39 |
| Built-up land | 21.65 | 3.90 | 8.09 | 13.89 | 0.00 | 0.05 |
| Unused land | 7.23 | 12.05 | 89.49 | 37.67 | −0.43 | 0.00 |

*4.2. Spatial Difference of Land Use Benefit*

It was found that the high values of land use benefits in the Yellow River Basin were concentrated in the nationally approved urban agglomerations such as the Central Plains, Guanzhong, hubao'e, and Shandong Peninsula. As the focus areas of economic development, high-density population gathering areas, and key areas in the comprehensive treatment of environmental pollution and ecological protection [52], these urban agglomerations had a large proportion of cropland and built-up land, rapid economic development, dense population, and industry [53]. Their technological development levels were also higher than that of other regions. Technological progress can improve the efficiency of resource utilization and the output capacity in the process of land use [54]. Meanwhile, the effect of population aggregation in the process of labor flow was gradually enhanced, which highlights the comparative advantage of urban agglomeration in regional development. Therefore, the input-output efficiency of land use in this region is high. Most of the land in the West and North was grassland and unused land. The population is sparse, the development is relatively backward, and the input and output value of land use is low. So, to narrow land use benefits' spatial differences between regions, urban agglomeration should give play to the radiation effect, strengthen the cooperation between regions, and drive the development of surrounding small and medium-sized cities. The spatial difference of ecological benefits in the Yellow River Basin was not obvious. The ecological benefits were at low levels in the whole country [55]. In recent years, the Yellow River Basin has made breakthrough progress in ecological construction and environmental governance [56]. However, the pressure on the ecological environment is still huge. Therefore, the key ecological environmental problems should be solved and the ecological compensation system should be improved actively in the treatment of the Yellow River Basin [57]. This would comprehensively control the interest relationship of various related factors in the process of development, protection and improvement of the Yellow River Basin, as well as take the road of the ecological economy.

*4.3. Agglomeration Features of Land Use Benefit*

The results of layer stacking showed that there was a certain spatial agglomeration of land use benefits in the Yellow River Basin. The revelation of the intra-regional correlation pattern and the reflection of the agglomeration characteristics of socio-economic phenomena often use the Getis-Ord Gi * [58]. The hot spots were mainly located in Shandong, Henan, Shaanxi, and Shanxi provinces in the east of the basin, as well as Lanxi urban agglomeration in the southwest and hubao'e urban agglomeration in the north (Figure 11). The cold spots were mainly distributed in Sichuan and Qinghai provinces in the West and Inner Mongolia Autonomous Region in the north. In 1995, the spatial agglomeration characteristics of cold spots and hot spots were particularly obvious, and the proportion of transition areas was relatively small. In 2018, the cold spots and hot spots of land use benefit decreased, but the transition areas increased. In addition, the distribution of cold spots and hot spots changed from centralized distribution to decentralized distribution. The change over the past 20 years further showed that the agglomeration characteristics of land use benefits in the basin had been gradually weakening. In addition, the land

use benefits had been becoming increasingly balanced. The main reasons are the imple-mentation of land use policy and the change of economic development mode [59]. For example, the implementation of the Central China Rise Strategy in 2006 and the Western Development Strategy in 2009 led to the gradual evolution of low-value areas of land use benefit into high-value areas in many central and western regions, effectively promoting balanced development of the spatial pattern of land use benefit. What's more, with the advancement of economic transformation, land use policies need to be adjusted to adapt to the new model of urban and rural development [60].

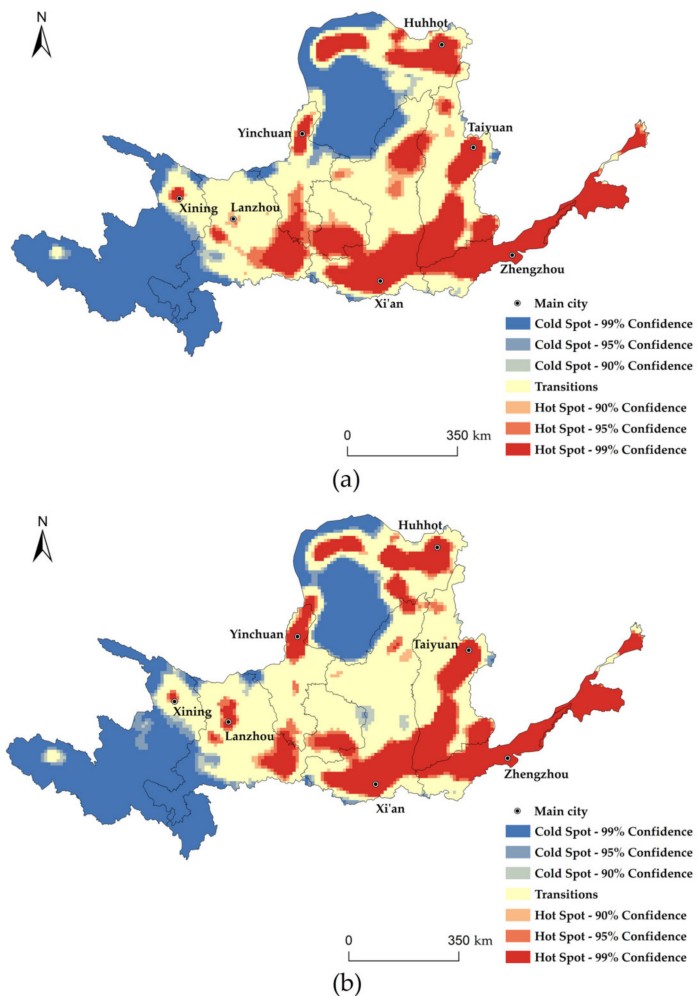

**Figure 11.** Hot spots distribution of land use benefits in the Yellow River Basin: (**a**) 1995; (**b**) 2018.

### 4.4. Implication and Limitation

The Yellow River basin is an important ecological barrier and economic zone in China, which is of great significance for maintaining social stability and ecological security, promoting ethnic unity and economic and social development of the country [61]. Many scholars have studied its development from the aspects of economic development [62] and ecological environment [63,64], but few have paid attention to its land use benefit. The Yellow River basin was relatively rich in land resources (Table 8). The main land use type was grassland, accounting for 47.58% of the total land area, followed by cropland, woodland, and unused land, accounting for 26.72%, 12.88%, and 8.60%, respectively. However, it was constrained by factors such as fragile natural ecological background and lack of water resources. Land development and utilization were greatly restricted. How to use land efficiently is important for the high-quality development of these watersheds. This study took the Yellow River Basin as the research object and enriched the relevant research

results. There were still some deficiencies in this study. First, because of the difficulty of obtaining data before 1995, the land use benefits before the reform and opening-up were not calculated. Secondly, due to the limitation of some data formats, many other indicators cannot be brought into the calculation. Finally, this study mainly considered the impact of land cover, without in-depth identification and analysis of the other driving factors causing benefit changes. So, it is still a research direction worthy of effort in the future.

**Table 8.** Proportion of land use area in the Yellow River Basin from 1995 to 2018.

| Land Use Types | Land Area Ratios/% | | | | | |
|:---:|:---:|:---:|:---:|:---:|:---:|:---:|
| | 1995 | 2000 | 2005 | 2010 | 2015 | 2018 |
| Cropland | 27.20 | 27.24 | 26.78 | 26.69 | 26.56 | 25.87 |
| Woodland | 12.69 | 12.70 | 12.91 | 13.00 | 13.02 | 12.96 |
| Grassland | 47.72 | 47.55 | 47.32 | 47.34 | 47.22 | 48.34 |
| Waters | 1.64 | 1.62 | 1.66 | 1.67 | 1.70 | 1.68 |
| Built-up land | 2.10 | 2.16 | 2.33 | 2.41 | 2.80 | 3.50 |
| Unused land | 8.65 | 8.73 | 9.00 | 8.89 | 8.70 | 7.65 |

## 5. Conclusions

Based on the grid-scale, this study selected the corresponding indicators to evaluate the land use benefit in the Yellow River Basin from 1995 to 2018, analyzing its space-time evolution law. During the study period, the land use comprehensive benefit, land use economic benefit, and land use social benefit in the Yellow River Basin all showed upward trends. Land use ecological benefit showed a rolling increase trend. Notably, land circulation had a certain impact on land use benefit. Therefore, it is very important to formulate reasonable and scientific land transfer policies.

There were obvious spatial differences in land use benefits. The spatial distribution pattern of comprehensive benefits was higher on the outside than in the middle, higher in the south than in the north, and higher in the east than in the west. Economic and social benefit were basically consistent with the distribution of comprehensive benefits, and the high-value areas were mostly located in the eastern and northern boundaries. Besides, the ecological benefit was basically in the middle area, and the distribution ranges of low and high values were small. The center of gravities of the three benefits were in the central region in the Yellow River Basin. Overall, the center of gravities of land use benefits had gradually moved north from 1995 to 2018. This result reflected that the governance of the Yellow River and the development and construction of poverty-stricken areas in the upper and middle reaches of the Yellow River achieved certain effects. In addition, the spatial distribution of land use benefits had a certain agglomeration effect. Urban groups generally belonged to high-value areas of land use benefit. The local governments should give full play to the leading roles of urban groups to narrow the development differences between regions.

**Author Contributions:** Conceptualization, Z.C. and F.L.; methodology, Z.C. and F.L.; software, Z.C., F.L. and J.S.; validation, Z.C., F.L. and Q.Z.; formal analysis, Z.C.; resources, F.L.; data curation, F.L.; writing—original draft preparation, Z.C.; writing—review and editing, F.L.; visualization, Z.C.; supervision, F.L.; project administration, Z.C. and F.L.; funding acquisition, F.L. All authors have read and agreed to the published version of the manuscript.

**Funding:** This research was financially supported by College Students Innovation and Entrepreneurship Training Program of China (No. S202010697111).

**Institutional Review Board Statement:** Not applicable.

**Informed Consent Statement:** Not applicable.

**Data Availability Statement:** http://www.resdc.cn (accessed on 15 June 2021), https://data.cnki.net/ (accessed on 15 June 2021).

**Acknowledgments:** We are thankful to the large number of scientists who have developed models and tools used to create the simulations used in this study. We also extend great gratitude to the anonymous reviewers and editors for their helpful reviews and critical comments.

**Conflicts of Interest:** The authors declare no conflict of interest.

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
