# Peer review of "Comprehensive Evaluation of Land Use Benefit in the Yellow River Basin from 1995 to 2018"

_land, doi:10.3390/land10060643_

Round 1

Reviewer 1 Report

The topic is interesting from the point of view of the land use benefit.

The authors contributed new approach to the land productivity, production effectivenes, economic and social benefits of land use. Interestingly, these benefits do not go hand in hand with the benefits of broadly understood ecology.

There are not many articles on this topic, therefore information with reference to presented example on this subject is valuable.

The manuscript is well written and the following sections are readable.

Despite this, I have two comments:

  1. Introduction:

After defining the purpose of research, it would be good that authors give clear research questions as an introduction to further stages. Clear research assumptions should follow from this.

  1. Material and Methods,

Study area: It would be good to add a general map with a location in relation to the global and national scale in the manuscript. It would be useful for the reader to orientate the research area in a broader geographical sense.

Author Response

Thank you for your suggestion. We have revised the article according to your advice.

Reviewer 2 Report

The article Comprehensive Evaluation and Temporal and Spatial Charac-2teristics of Land Use Benefit in the Yellow River Basin is focus on calculating the land productivity in the Yellow River Basin including the ecological benefits and urban grow.

In order to be publishing some minor revision are necessary:

Q1. In Figure 1 it is hard to understand the map contents. If you put 2 in a row and not 3 as they are now the content would be more visible. The legend is also hard to read.

Q2. In section 2.3. Study methods the formula used for land use economic benefit, the land use social benefit, land use ecological benefit sunt formule preluat? Then they should be quoted. If they are own formulas, this aspect must be specified.

Q3. The same aspect must be done for formulas 8-16.

Q4. Figure 4. The maps are close to friends to observe their details. They should be enlarged, to add punctually some identification elements (ec. Cities). The same observation is valid for figures 6, 7,8.

Q5. Figure 11 although it is very important and highlights very well what are the hot spots mermaids is not legible. You should export maps at a minimum of 300 dpi.

Best regards

Author Response

Thanks for your advice. We have revised the article according to your suggestion.

Reviewer 3 Report

Main comments

Title. The title could be shortened or changed to "Changes in land use benefits in the Yellow River Basin from 1995 to 2018"  or  "Assessing changes in land use benefits in the Yellow River Basin from 1995 to 2018", this one stressing the methodology issues as well.

English language.  The authors should accommodate verb tenses to better convey some ideas and review English Style. I know that tenses in Chinese are much simpler, but a quick review should help to change some sentences written in the past tense to present tense for better flow. Eg. "The Yellow River Basin (Figure 1) originated in Bayan Har Mountain... " should be changed to "The Yellow River Basin (Figure 1) originates in Bayan Har Mountain ...". There are some free tools such as Grammarly which can help with this. 

Line 226. "The Jenks was used to divide land use benefit into low, medium, and high-value areas " This is a methodological sentence and should be moved to the corresponding section.  Jenks actually refers to the Jenks optimization method or Jenks natural breaks classification method. The citation can be  Jenks, G. F. 1967. "The Data Model Concept in Statistical Mapping", International Yearbook of Cartography 7: 186–190; or this one JENKS, G.F. and CASPALL, F.C. (1971), ERROR ON CHOROPLETHIC MAPS: DEFINITION, MEASUREMENT, REDUCTION. Annals of the Association of American Geographers, 61: 217-244. https://doi.org/10.1111/j.1467-8306.1971.tb00779.x

Line 262. "The result of Layer Stacking ..." This is not described in the Methods section. Authors can add a paragraph explaining this after describing the methods to calculate each benefit. 

Line 277. This sentence also needs to be included in the Methods section: "The movement of the geographical center of gravity can reflect the change of geographical elements. " as it explains how the result of this analysis will be interpreted.  And they could also change the last part to "the change in dominance of the land use benefits obtained" or something along this lines.

Line 278. This sentence could be omitted or rephrased "Therefore, the change in  high-density area of benefit was discussed through the analysis on the center of gravity of land use benefit trajectory in the Yellow River Basin".

Line 298. Figure 10. An overall evolution graph as used in Figure 9 would be better, or alternatively, this figure could be completely omitted. 

Tables  5 to 7. These tables should be included in the Results section. 

Line 366.  Agglomeration features of land use benefit. The use of  Getis-Ord Gi * hot-spot analysis should be described in the Methods section.  

Author Response

(The authors gave the same response as above.)
